# Bioaccessibility, Intestinal Absorption and Anti-Inflammatory Activity of Curcuminoids Incorporated in Avocado, Sunflower, and Linseed Beeswax Oleogels

**DOI:** 10.3390/foods13030373

**Published:** 2024-01-24

**Authors:** Patricia Ramírez-Carrasco, Ailén Alemán, Estefanía González, M. Carmen Gómez-Guillén, Paz Robert, Begoña Giménez

**Affiliations:** 1Department of Food Science and Chemical Technology, Faculty of Chemical and Pharmaceutical Sciences, University of Chile, Santos Dumont 964, Independencia 8380494, Santiago, Chile; patricia.ramirez@ug.uchile.cl; 2Institute of Food Science, Technology and Nutrition (ICTAN-CSIC), José Antonio Nováis 10, 28040 Madrid, Spain; ailen@ictan.csic.es (A.A.); cgomez@ictan.csic.es (M.C.G.-G.); 3School of Health, Universidad de O’Higgins, Av. Libertador Bernardo O’Higgins 611, Rancagua 2820000, Cachapoal, Chile; estefania.gonzalez@uoh.cl; 4Department of Food Science and Technology, Faculty of Technology, University of Santiago of Chile, Av. Víctor Jara 3769, Estación Central 9170124, Santiago, Chile

**Keywords:** beeswax oleogels, curcuminoids, in vitro digestion, oxidative stability, intestinal absorption, anti-inflammatory properties

## Abstract

Beeswax oleogels (OGs), with a mechanical strength similar to pork backfat, were formulated with avocado (A), sunflower (S), and linseed (L) oils, applying a central composite design plus star point, and were evaluated as oral delivery vehicles of curcuminoids (OGACur, OGSCur, OGLCur). The incorporation of curcumin into the OG matrix significantly delayed both the formation of peroxides and conjugated trienes (K_268_ values), and the degradation rate of curcumin decreased with the increase of the oil polyunsaturated fatty acids (PUFA) content. The oil structuring did not affect the bioaccessibility of curcuminoids (>55% in all the OGs, regardless of the oil type), but it did reduce the release of fatty acids (~10%) during in vitro gastrointestinal digestion. The intestinal absorption, evaluated in Caco-2 cell monolayers, was higher for the micelle-solubilized curcumin from the digested OG than from unstructured oils, and it showed high anti-inflammatory potential by inhibiting the tumor necrosis factor-α (TNF-α) production compared to the positive control, both before and after the stimulation of ThP-1 cells with LPS. Regardless of the oil type, these beeswax-based OGs with gel-like behavior designed as fat replacers may be promising vehicles for the oral delivery of curcuminoids.

## 1. Introduction

Oleogels (OGs) are semi-solid systems, in which the oil is entrapped within a three-dimensional network formed by an oleogelator [1]. Oleogelators can be classified into two main groups according to their molecular weight: high molecular weight oleogelators that form three-dimensional polymeric networks that immobilize a large volume of oil; and low molecular weight oleogelators (LMOGs), which self-assemble into crystal networks, entrapping the oil [2]. Natural waxes emerged as promising LMOGs due to their availability and reasonable cost. Among natural waxes, beeswax, recognized as Generally Recognized as Safe (GRAS), has been widely used for its capacity to structure vegetable oils at low concentrations, showing better structuring performance than other vegetable waxes, such as rice bran or sunflower wax [3].

Interest in OGs has considerably increased in recent years, driven by their potential as substitutes for animal fats in a variety of foods, including spreads, bakery, confectionary, dairy, and meat products. This serves as a strategic approach to formulate foods with reduced saturated and *trans* fats [1,3,4]. In such applications, OGs are formulated with oils rich in monounsaturated fatty acids (MUFA) and/or polyunsaturated fatty acids (PUFA), such as linseed, sesame, corn, canola, safflower, soybean, sunflower, olive, or fish oils [3,5]. The selection of the oil, together with the type and concentration of organogelator and the processing conditions, stands as an important factor in OG formulation, impacting their rheological, textural, and thermal properties [1,5,6,7]. In this study, three oils (linseed, sunflower, and avocado oil) were chosen for their different content of saturated fatty acids (SFA), MUFA, and PUFA. Additionally, the utilization of avocado oil in OG formulation remains largely unexplored. 

The predominant approach for obtaining OGs involves the direct dispersion of the oleogelator into the oil, leading to the formation of crystallite or self-assembled networks [8]. This process involves continuous agitation of the oil–oleogelator blend at temperatures surpassing the melting point of the oleogelators. However, this process may induce lipid oxidation of the unsaturated fatty acid-rich oils utilized in OG formulation. To counteract this, the incorporation of lipophilic antioxidants into the OG, while still scarce, emerges as a potentially effective strategy to mitigate lipid oxidation during both the OG formulation and storage. Curcumin, the major polyphenol found in turmeric rhizomes, has been incorporated in various studies to enhance the oxidative stability of PUFA-rich OGs [9,10,11,12]. Curcumin exhibits diverse biological activities, including antioxidant, anti-inflammatory, antiviral, and antitumor effects [13]. However, its practical application is constrained due to its low water solubility, chemical instability, and fast metabolism, significantly impacting its bioaccessibility and oral bioavailability [14]. To address these limitations, enhancing the solubility of curcumin through the development of delivery systems, such as lipid-based vehicles, is a strategy that has received increasing attention in recent years [14].

More recently, OGs have emerged as promising delivery systems for lipophilic bioactive compounds, including D-limonene, quercetin, hesperidin, capsaicin, β-carotene, lutein, or curcuminoids. This is attributed to their capability to dissolve, stabilize, and deliver such bioactive compounds effectively [15,16]. The OG network serves as a protective barrier against the degradation of bioactive compounds within the gastrointestinal tract. However, upon digestion by intestinal lipases, the resulting mixed micelles can solubilize the lipophilic bioactives, enhancing their bioaccessibility and bioavailability [15]. Despite this potential, the design of OGs as vehicles for delivering bioactive compounds remains limited, with only a few studies in the literature evaluating the bioaccessibility and/or bioavailability of incorporated bioactive compounds after a simulated gastrointestinal digestion [10,14,16,17,18,19,20]. However, this aspect is crucial, directly impacting the performance of OGs as delivery systems in the human gut. The nature of the oleogelator has been reported to influence the bioaccessibility of the lipophilic compounds incorporated in OGs [16,17,20]. Therefore, it is essential to assess the role of each oleogelator when used for the formulation of OGs as delivery systems for bioactive compounds. In this context, the novelty of this study lies in the comprehensive evaluation of the bioaccessibility, bioavailability, and anti-inflammatory properties of curcumin beeswax OGs formulated with avocado, sunflower, and linseed oils, and mechanical strength similar to pork backfat, after in vitro simulated gastrointestinal digestion.

This study aimed to provide insights into the relationship between oil type in beeswax OG formulations and the bioaccessibility, intestinal absorption, anti-inflammatory properties, and oxidative stability of curcumin. The findings will contribute to a better understanding of the potential of beeswax OG formulations as delivery vehicles for curcumin in terms of their oil type-dependent effects.

## 2. Materials and Methods

### 2.1. Materials

Linseed oil (unrefined cold-pressed oil) was purchased from Nutra Andes Ltda. (Valparaiso, Chile), extra virgin avocado oil (unrefined cold-pressed oil) was purchased from Chilean Taste FOOD (Peumo, Chile), and sunflower oil (refined, bleached, deodorized oil) was purchased from a local market (Santiago, Chile). The fatty acid profile of these oils, expressed as the percentage of methyl esters, is shown in the Appendix A. Beeswax was obtained from Coprin S.A. (Santiago, Chile), and curcumin (79.3% curcumin, 18.2% demethoxycurcumin, 2.6% bisdemethoxycurcumin) was purchased from Xi’an Xin Sheng Bio-Chem Co. (Xi’an, China). Pepsin from porcine gastric mucosa (P7012, 2500 AU/mg), pancreatin from porcine pancreas (P7545, 8 x USP specifications), and porcine bile extract (B8631) were purchased from Sigma-Aldrich (Santiago, Chile). All other reagents were purchased from Sigma-Aldrich (Santiago, Chile), and analytical HPLC and GC-grade solvents were obtained from Merck S.A. (Santiago, Chile). Human colon adenocarcinoma cells (CaCo-2) and the human leukemia monocytic cell line (ThP-1) were obtained from the cell bank of the Centro de Investigaciones Biológicas Margarita Salas, CSIC (Madrid, Spain). The CCK-8 cell counting kit was purchased from Sigma-Aldrich (Madrid, Spain), and the ELISA kit was purchased from Diaclone (Besançon Cedex, France). MEM-alpha medium, Roswell Park Memorial Institute (RPMI) 1640 medium, fetal bovine serum, streptomycin/penicillin, and Phorbol 12-myristate 13-acetate (PMA) were purchased from Gibco, Thermo Fisher Scientific (Waltham, MA, USA). 

### 2.2. Methods

#### 2.2.1. Preparation of Oleogels

The beeswax concentrations used to formulate the OGs in this study (9.12%, 8.24%, and 7.92% for OGLCur, OGACur and OGSCur, respectively) were defined in previous studies that applied experimental designs (central composite design plus star point), in which the beeswax and curcumin concentrations were the independent variables and the mechanical strength and oil binding capacity (OBC) were the response variables. The desirability function was used for multiple response optimization (Appendix A), in which OBC was maximized, and the mechanical strength was fixed to that obtained for pork backfat by back extrusion in a previous study (18.4 ± 0.5 N) [11]. The OGs (100 g) with curcumin (0.2% *w*/*w*, Cur) were formulated using linseed (L), sunflower (S), or avocado (A) oils as follows: beeswax (9.12 g in OGLCur, 7.92 g in OGSCur, or 8.24 g in OGACur) was dispersed in oil (90.88 g L, 92.08 g S, or 91.76 g A) and heated at 70 °C (over the melting temperature of beeswax [11]) under stirring (700 rpm). Curcumin (200 mg) was previously incorporated into an aliquot of oil (20 g) and heated to 140 °C [17] to guarantee complete dissolution. When the temperature was reached, curcumin in oil was added to the beeswax–oil mixture and stirred for 1 min. The resulting solution was held at room temperature for 30 min and subsequently stored at 4 °C until analysis. 

#### 2.2.2. Oleogel Characterization

##### 2.2.2.1. Mechanical Strength

The mechanical strength of the OGs was measured by back extrusion, employing a texturometer (BDO-FDO.5T5, Zwick/Roell, Ulm, Germany) fitted with a 5 kg load cell and a stainless-steel cylindrical probe (18 mm diameter). The OGs were penetrated up to 30 mm at a rate of 1.5 mm/s. Mechanical strength (N) was expressed as the average force in the last quarter of the test. 

##### 2.2.2.2. Oil Binding Capacity

Oleogels were cut into cylindrical pieces (1 cm × 1 cm), weighed, and positioned on filter papers (Whatman #1) supported by a Petri plate. Loss of oil was determined by weighing the filter papers and Petri plates before placing the sample and after 1 day and 7 days of storage at 25 °C. The OBC was calculated using Equation (1):
(1)
$$OBC\left(\%\right)=100-\frac{W_{tf}-W_{t0}}{W_{s}}\times 100$$

where W_tf_ is the weight (g) of the Petri dish and filter paper after the storage time (1 and 7 days), W_t0_ is the weight (g) of the dish with filter paper without the sample at the beginning of storage, and W_s_ is the weight (g) of the OG sample.

##### 2.2.2.3. Rheological Analysis

Rheological properties of the OGs were assessed using a rheometer (HR-2 Discovery Hybrid Rheometer, TA Instruments, New Castle, DE, USA) with a sandblasted parallel-plate geometry (40 mm diameter, 1 mm gap). Shear strain (γ) varied from 0.01% to 20% at 6.28 rad/s and 25 °C to determine the linear viscoelastic region. Frequency sweeps were performed from 0.1 to 100 rad/s, maintaining a temperature of 25 °C and a shear strain (γ) of 0.02% (within the linear viscoelastic region).

#### 2.2.3. Oxidative Stability of Oleogels under Accelerated Conditions

Melted OGs (4 g) were placed in glass tubes with screw caps (12 mL) in an oven at 60 ± 1 °C (IPP100, Memmert, Germany). The tubes were periodically removed by triplicate to determine peroxide value (PV) [21] and K_268_ values (conjugated trienes) [22]. The PV was expressed as mEqO_2_/kg oil according to the following Equation (2):
(2)
$$PV=\frac{\left(V_{s}-V_{b}\right)\times N\times 1000}{W_{s}}$$

where V_s_ and V_b_ are the volume of sodium thiosulfate to titrate the OG sample and blank, respectively; N is the normality of the sodium thiosulfate; and W_s_ is the weight of the OG sample.

For the determination of K_268_ values, 20 mg of OG was dissolved in isooctane to 25 mL. A UV-Vis spectrophotometer (Orion Aquamate 8000, Thermo Fisher Scientific, Waltham, MA, USA) was used to measure absorbance at 268 nm. The results were expressed as K_268_ values, according to the following Equation (3):
(3)
$$K_{268}=\frac{\mathrm{Absorbance}\mathrm{at}268\mathrm{nm}}{C}$$

where C is the oil concentration (g/100 mL) in the isooctane solution.

Furthermore, the curcumin content was determined during the storage time. Curcumin was extracted from the OG matrix (0.5 g) with methanol (15 mL). The mixture was heated at 80 °C in a water bath until melting, vortexed (30 s), and centrifuged (7500× *g*, 4 °C, 10 min). The supernatant was collected, and two more successive extractions were performed from the pellets. All the supernatants were mixed and transferred to a volumetric flask (50 mL). An aliquot was injected into the HPLC to determine curcumin according to Marczylo et al. [23]. Data were fit to a first-order kinetic model according to Equation (4):
(4)
$$C=C_{0}\times e^{-kt}$$

where C_0_ and C are the initial content and the content of curcumin at time t in the OG, respectively, and k is the first-order degradation rate constant.

#### 2.2.4. In Vitro Gastrointestinal Digestion of Oleogels

Oleogel samples were subjected to in vitro gastrointestinal digestion according to Brodkorb et al. [24], simulating oral, gastric, and intestinal phase conditions. Briefly, 0.5 g of OG was finely chopped and dispersed in distilled water up to 5 mL. Afterwards, simulated salivary solution (4 mL), 0.3 M CaCl_2_ (25 µL) and distilled water (975 µL) were added to the dispersion and incubated at 37 °C and 170 rpm for 2 min. Subsequently, the oral bolus (10 mL) was mixed with the simulated gastric solution (8 mL), 0.3 M CaCl_2_ (5 µL), pepsin (2000 U/mL gastric phase), and HCl (0.1 M) to adjust the pH to 2.0 and distilled water up to 20 mL. The final volume was incubated at 37 °C and 170 rpm for 120 min. Finally, the gastric bolus (20 mL) was mixed with the simulated intestinal solution (16 mL), 0.3 M CaCl_2_ (40 µL) and NaOH (1 M) to adjust the pH to 7.0, bile extract (10 mM), and pancreatin (2000 U lipase activity/mL intestinal phase). A pH-stat (902 Titrando, Metrohm, Switzerland) was employed to sustain a pH of 7.0 for 2 h at 37 °C through the continuous addition of NaOH (1 M) under constant stirring. The digested samples were centrifuged at 50,000× *g* and 4 °C for 45 min, and the aqueous phase (micellar phase) was collected and stored at −80 °C until analysis.

##### 2.2.4.1. Bioaccessibility of Total FFA, Individual FFA, and Curcumin from Digested Samples

The extraction of free fatty acids (FFAs) released during the intestinal digestion phase was performed according to Ng et al. [25]. The FFAs were methylated with H_2_S_2_O_4_ in methanol (0.06%, *v*/*v*), and the fatty acid methyl esters (FAMEs) were analyzed using a gas chromatograph (7890B, Agilent Technologies, Santa Clara, CA, USA), with a flame ionization detector and HP-88 column (fused silica capillary column 0.25 mm inner diameter, 0.20 μm film thickness, 100 m, Agilent Technologies, USA) according to Álvarez et al. [26]. The identification and quantification of individual fatty acids were accomplished using a calibration curve derived from FAME external standards (C16:0, 0.05–2.58 mg/mL; C16:1, 0.01–1.04 mg/mL; C18:0, 0.01–1.0 mg/mL; C18:1, 0.02–8.48 mg/mL; C18:2, 0.01–8.48 mg/mL; C18:3, 0.01–18.22 mg/mL; R^2^ > 0.99 for all of them). 

The extraction of curcumin from the micellar phase was performed with isopropanol as described by Calligaris et al. (2020) and was quantified using an HPLC (UltiMate 3000, Thermo Scientific, Waltham, MA, USA) coupled to a UV/VIS detector (VWD-3100) and an Acquity BEH shield RP18 column (130Å, 2.1 mm × 100 mm, 1.7 µm; Waters, Milford, CT, USA) according to Marczylo et al. [23]. The quantification of curcuminoids was performed at 425 nm, using calibration curves for curcumin (0.005–0.125 mg/mL; R^2^ = 0.99), demethoxycurcumin (0.002–0.150 mg/mL; R^2^ = 0.99), and bisdemethoxycurcumin (0.00025–0.05 mg/mL; R^2^ = 0.99). A typical HPLC chromatogram of curcuminoids was included in the Appendix A.

The bioaccessibility of every major fatty acid and curcumin were calculated according to Equation (5):
(5)
$$Bioaccessibility(\%)=\frac{\mathrm{mg}\mathrm{of}\mathrm{FFA}\mathrm{or}\mathrm{curcuminoids}\mathrm{in}\mathrm{micellar}\mathrm{phase}}{\mathrm{mg}\mathrm{of}\mathrm{FA}\mathrm{or}\mathrm{curcuminoids}\mathrm{in}\mathrm{undigested}\mathrm{sample}}\times 100$$


#### 2.2.5. Cell Model Assays

##### 2.2.5.1. Cell Viability

CaCo-2 and ThP-1 cell viability were determined as described by Alemán et al. [27]. Cells (90 µL) were seeded in 96-well culture plates (1 × 105 cells/mL) and incubated (37 °C, 5% CO_2_) until 80–90% confluence was reached. ThP-1 cells (2 × 10^5^ cells/mL) were differentiated by adding 25 nM PMa, incubated for 72 h until attachment was achieved, and allowed to settle for 24 h in RPMI medium. Digested samples (10 µL) were added into the wells to both cell lines (undiluted for CaCo-2 and 1:25 diluted for ThP-1). The incubation time (37 °C, 5% CO_2_) to which each cell line was exposed (4 h for CaCo-2 or 24 h for ThP-1) was determined based on the duration of subsequent studies. Cell viability was evaluated using the cell counting kit-8 (CCK-8) according to manufacturing instructions.

##### 2.2.5.2. CaCo-2 Cell Permeability Assay 

CaCo-2 cells were seeded in Transwell inserts (2 × 10^5^ cells/cm^2^) with a semi-permeable polycarbonate membrane (12 mm diameter, 0.4 µm pore size, Corning Costar, Cambridge, MA, USA) [27]. Monolayers were formed after 21 days of incubation (37 °C, 5% CO_2_), and their integrity was assessed by measuring transepithelial/transendothelial electrical resistance (TEER) with Millicell-ERS equipment (Millipore, Burlington, MA, USA). Monolayers exhibiting TEER values over 350 Ohms were employed for the experiment. To assess the apical-to-basolateral (AP-BL) permeability of the digested samples, 1.5 mL of fresh culture medium was added to the BL side, and 0.5 mL of digested sample dissolved in fresh culture medium (1:10) was applied to the AP side. Following incubation at 37 °C with 5% CO_2_, 0.5 mL of the BL solution was collected at 1 and 2 h. The percentage of curcumin that passed through the cell monolayer (relative to the initial curcumin amount) was determined using HPLC as outlined in Section 2.2.4.1. 

The apparent permeability coefficients (P_app_) were calculated according to Equation (6):
(6)
$$P_{app}=\frac{dQ}{dt}\times \frac{1}{A\times C_{0}}$$

where dQ/dt is the curcumin flux (μg/s) across the barrier, A is the surface area of the inserts (cm^2^), and C_0_ is the initial curcumin concentration on the AP side (μg/mL).

##### 2.2.5.3. Immune Stimulation Assay

The measurement of the immune response of ThP-1 cells differentiated to macrophages was performed in 24-well plates by seeding 2 × 10^5^ cells/well (900 µL) [27]. For the study, one set of plates with ThP-1 was stimulated with lipopolysaccharide (LPS, 1 µg/mL) for 4 h and then incubated (18 h at 37 °C and 5% CO_2_) with 100 µL of each digested sample (1:25) to evaluate their ability to regulate the inflammatory response once initiated. On the other hand, 100 µL of digested sample (1:25) was firstly added to the other set of plates with ThP-1, incubated for 18 h at 37 °C and 5% CO_2_, and then stimulated with LPS (1 µg/mL) to determine their protective anti-inflammatory activity. Positive (untreated-stimulated cells) and negative (untreated-unstimulated cells) controls were used to verify the inflammatory effect. After the different treatments, samples were centrifuged (230× *g*, 5 min). The concentration of tumor necrosis factor-α (TNF-α) cytokine released in the supernatants was quantified using an ELISA reaction kit according to the manufacturing instructions.

#### 2.2.6. Statistical Analysis

All the experiments were conducted in triplicate. The analysis of variance (ANOVA) and Tukey’s multiple range test were employed to identify statistically significant differences (*p* ≤ 0.05) among samples. Statgraphics Centurion 15.1 software (Stat Technologies Inc., Golden Valley, MN, USA) was utilized for these statistical evaluations.

## 3. Results

### 3.1. Oleogel Characterization

Table 1 shows the mechanical strength and OBC of the OGs containing curcumin. The mechanical strength values were 19.5 ± 1.2 N, 18.9 ± 0.4 N, and 18.5 ± 1.6 N for OGLCur, OGSCur, and OGACur, respectively. All the OGs exhibited similar (*p* > 0.05) mechanical strength values, close to that of the pork backfat used as the target (18.4 ± 0.5 N) [11]. However, achieving this target value required slightly different concentrations of beeswax depending on the oil type (9.12%, 8.24%, and 7.92% for OGLCur, OGACur, and OGSCur, respectively). This variation may be linked to the solubility of beeswax in the oils. The slightly higher beeswax concentration in OGLCur needed to meet the target value could be attributed to the higher content of PUFA and lower levels of SFA and MUFA in linseed oil compared to sunflower and avocado oils (Appendix A), since a higher rate of low melting triglycerides has been related with the increased solubility of wax components in the oil [5,28,29], potentially influencing the type of molecular interactions occurring during OG crystallization [6]. Beeswax concentrations were comparable in OGACur and OGSCur, despite sunflower oil having a higher content of low melting triglycerides, with higher percentages of PUFA and lower contents of SFA and MUFA than the avocado oil (Appendix A). This result may be attributed to the higher content of polar minor components that could be expected in the unrefined cold-pressed avocado oil used in this study, likely contributing to an increase in the bulk polarity of this oil [30]. Therefore, the gelling ability of oleogelators, such as beeswax, is influenced not only by the fatty acid composition but also by other factors, such as the molecular weight, the chain length of fatty acids, and the presence of minor components in edible oils [1,4,5].

OBC is associated with the ability of the crystalline structure of beeswax to retain oil. High OBC values (≥86%) were observed in all the OG samples on day 1 of storage (Table 1). A significant decrease in OBC (*p* ≤ 0.05; 11–14%) occurred in the three OGs after 7 days of storage. The most substantial oil loss occurred rapidly during the first day of storage. Subsequently, while oil continued to migrate out of the OGs, the rate of migration significantly slowed. These results are in agreement with the stages of oil loss reported for wax OGs, an initial rapid or short-term stage associated with unbound oil followed by a slower or long-term stage related to oil adsorbed on the surface of the beeswax particles [31].

The mechanical spectra resulting from frequency sweeps of the OG are shown in the Appendix A. Regardless of the oil type, all the OGs showed gel-like behavior, with elastic modulus (G′) values surpassing viscous modulus (G″) values across the entire frequency range studied. This characteristic makes them suitable as solid fat replacers, while reducing the content of SFA and *trans* fatty acids [1]. Both G′ and G″ showed a low-frequency dependence in the studied range (0.1–100 rad/s) in all the OG samples, indicating the ability of the OG to withstand external forces within the specified frequency range. This gel-like behavior aligns with descriptions typically found in studies of wax-based OGs [6,16,20].

### 3.2. Oxidative Stability of Oleogels under Accelerated Conditions

The evolution of PV in OGs with curcumin (OGLCur, OGSCur, OGACur) and OGs without curcumin (OGL, OGS, OGA), used as controls during storage at 60 °C, is shown in Figure 1a–c. 

All OGs showed both the formation and decomposition of peroxides throughout the storage. The control OGs reached their maximum PV at 25, 80, and 120 days for OGL (38 mEqO_2_/kg oil), OGS (120 mEqO_2_/kg oil), and OGA (126 mEqO_2_/kg oil), respectively. The PV evolution correlated with the degree of oil unsaturation, with OGL reaching the maximum PV in the shortest storage time. Moreover, the maximum PV reached by OGL was lower than in OGS and OGA. In this context, Lim et al. [32] reported a PV close to 120 mEqO_2_/kg oil in canola beeswax OG stored at 60 °C, similar to the maximum PV found in OGS and OGA. However, Hwang et al. [33] reported a maximum PV close to 45 mEqO_2_/kg oil in fish oil beeswax OG stored at 50 °C, resembling the maximum PV found in OGL. This suggests that lower PV is associated with highly unsaturated systems, in which peroxide decomposition occurs faster than formation, resulting in a lower maximum PV. As expected, the inclusion of curcumin (0.2% *w*/*w*) in OGs significantly delayed peroxide formation. The main antioxidant mechanism of curcumin in lipid matrices involves hydrogen atom donation from its hydroxyl groups [34]. Li et al. [10] reported that curcumin incorporation increased the oxidative stability of corn oil OG structured with β-sitosterol and lecithin, reducing both the PV and *p*-anisidine value, although the storage time at 60 °C was shorter in that study (10 days). Secondary oxidation products, measured by K_268_ values representing conjugated trienes, exhibited significant increases (*p* ≤ 0.05) during storage for both control OGs and curcumin OGs. However, OGLCur, OGSCur, and OGACur showed lower values than the control OGs, attributed to the antioxidant activity of curcumin.

Moreover, the degradation kinetics of curcumin were determined in the OGs during storage. Figure 1d shows the degradation rate constant (*k*) of curcumin in OGLCur, OGSCur, and OGACur vs. the percentage of PUFA in the linseed, sunflower, and avocado oils used for the OG formulation. OGACur showed the highest *k* value (0.0055 ± 0.0002 days^−1^; R^2^ = 0.905), followed by OGSCur (0.0026 ± 0.0001 days^−1^; R^2^ = 0.810), and OGLCur (0.0020 ± 0.0001 days^−1^; R^2^ = 0.916). This indicates that curcumin was the most susceptible substrate to oxidation in OGACur, given the low content of PUFA in avocado oil (13.75%, Appendix A). Conversely, PUFAs (48.46% and 71.53% in OGSCur and OGLCur, respectively; Appendix A) were the most susceptible substrates to oxidation in OGSCur and OGLCur; therefore, the degradation rate of curcumin was lower in these matrices. These results suggest that the degree of oil unsaturation may influence the degradation rate of curcumin when incorporated into lipophilic matrices, with higher degradation rates in oils with low degrees of unsaturation. A similar behavior was found for the tocopherol stability in oils with different unsaturation levels, where faster tocopherol loss was found in monounsaturated oils, while polyunsaturated oils retained significant amounts of tocopherol [35]. 

Although the accelerated oxidation study of the OGs with and without curcumin was conducted at 60 °C for comparative purposes with other studies [10], this storage temperature exceeded the melting temperature of the OGs (Appendix A), at which the OGs exhibited liquid-like viscous behavior, similar to liquid oils. Both lipid oxidation and curcumin degradation would be expected to slow down at storage temperatures under the melting point due to the formation of the crystalline network immobilizing the oil, which in turn reduces the oxygen diffusion coefficient and restricts the movement of reactants [3].

### 3.3. Bioaccessibility of Individual FFAs and Curcumin from Digested Samples

The lipolysis of both unstructured oils (LCur, SCur, ACur) and OGs (OGLCur, OGSCur, OGACur) was assessed by measuring the release of fatty acids (FFA) after the intestinal phase (Figure 2). All unstructured oils, regardless of the oil type, showed bioaccessibility values of total FFA close to 60% (*p* > 0.05). Similar results have been reported for unstructured sunflower and linseed oils [20,36]. The bioaccessibility of total FFA in OGs was around 10% lower (*p* ≤ 0.05) than in the corresponding unstructured oils, with values ranging from 46.1 to 53.5%. The type of oil used to formulate the OG did not have a significant effect on the release of fatty acids, and OGLCur, OGSCur, and OGACur showed similar extents of lipolysis (*p* > 0.05). Furthermore, the bioaccessibility of major individual FFAs was also significantly (*p* ≤ 0.05) lower in OGs than in unstructured oils (Figure 2). The type and gelation mechanism of the oil structuring agent employed in the OG formulation, the mechanical strength of the OGs, and the presence of several components in the matrix that may directly interfere with the digestion process can influence the extent of lipolysis. A sufficiently structured OG network can impede the access of lipases and bile salts to the liquid oil trapped in the OG matrix, thereby preventing lipid hydrolysis and emulsification and, consequently, reducing lipolysis. OGs with high mechanical strength have been associated with a lower extent of lipolysis in some studies in which ethylcellulose or phytosterol OGs were subjected to in vitro digestion [18,37,38]. This is attributed to the fact that harder OGs are more resistant to mechanical breakdown during digestion, leading to a slower release of oil that remains inaccessible to lipases. However, this relationship between mechanical strength and the extent of lipolysis is not universally consistent. Ashkar et al. [37] reported an inverse relationship between OG hardness and the extent of lipolysis, which was attributed to the contribution of the oil structuring agent (monoglycerides and diglycerides) in micelle formation during digestion.

Moreover, the bioaccessibility of major individual FFAs tended to decrease with an increase in the unsaturation degree of fatty acids in all the digested samples (Figure 2). This trend has also been reported after the in vitro digestion of encapsulated linseed oil, in which the bioaccessibility of FFAs decreased with an increase in both the chain length and the unsaturation degree [26]. In this context, the position of the fatty acid in the triglyceride, its chain length, and the degree of unsaturation have been reported to impact the extent of triglyceride lipolysis [36].

Table 2 shows the bioaccessibility of bisdemethoxycurcumin, demethoxycurcumin, and curcumin, as well as the bioaccessibility of total curcuminoids, after simulated gastrointestinal digestion of unstructured oils (LCur, SCur, ACur) and OGs (OGLCur, OGSCur, OGACur).

The bioaccessibility of total curcuminoids was mainly represented by the curcumin content, constituting approximately 80% of the total curcuminoids, in agreement with Calligaris et al. [20]. The oil type used in the OG formulation did not exert any significant effect on the bioaccessibility of curcumin or total curcuminoids, and all the OGs showed similar (*p* > 0.05) values, ranging from 53.7% to 59.7% for curcumin and 55.9% to 61.5% for total curcuminoids. Yi et al. [39] studied the impact of the unsaturation degree of oils on the bioaccessibility of β-carotene nanoemulsions, reporting comparable values in both MUFA-rich oils and PUFA-rich oils. Moreover, both OG and unstructured oils showed similar values (*p* > 0.05) for the bioaccessibility of total curcuminoids, bisdemethoxycurcumin, demethoxycurcumin, and curcumin. This suggests that oil structuring did not interfere with the release of curcuminoids transferred from the OG matrix to the micellar phase due to lipid micellation. Calligaris et al. [20] reported that the type of structuring agent may significantly affect the bioaccessibility of curcuminoids incorporated into OGs, decreasing to values of ~30% in the case of rice wax and monoglyceride OG. In contrast, phytosterol OG and unstructured sunflower oil showed a curcumin bioaccessibility of 45% and 50%, respectively, which were lower than the curcumin bioaccessibility found for beeswax OGs in this study (53.7 to 59.7%). In another study, Li et al. [10] reported that the bioaccessibility of curcumin in curcumin-loaded corn oil OG structured by β-sitosterol and lecithin (~67%) was twice as high as that of unstructured corn oil (~36%). This difference was attributed to the higher amounts of micelles formed due to the involvement of lecithin in their formation, leading to an increase in the micelle-solubilized curcumin.

Curcumin showed the lowest bioaccessibility compared to demethoxycurcumin and bisdemethoxycurcumin, which displayed the highest bioaccessibility (Table 2). This may be related to the number of methoxy groups in their structure and, consequently, the polarity of the compounds. Curcumin, with two methoxy groups, is the least polar compound, explaining its lower release from the OGs into the digestive fluids. This is followed by demethoxycurcumin, with one methoxy group, and bisdemethoxycurcumin, which is the most polar compound without any methoxy group. Additionally, curcumin has been reported to be more susceptible to oxidation under intestinal conditions than demethoxycurcumin and bisdemethoxycurcumin, which may contribute to the lower bioaccessibility values found for this isomer [20].

### 3.4. Cell Culture Assays

#### 3.4.1. Cell Viability

Cell viability results for both CaCo-2 and ThP-1 cell lines, following their incubation with the micellar phases of the digested unstructured oils (LCur, SCur, ACur) and OGs (OGLCur, OGSCur, OGACur), compared to a control of unexposed cells, are shown in the Appendix A. Cell viability over 95% was observed in CaCo-2 (Appendix A) after incubation with the 1:10 diluted micellar phases (4 h) of all the digested samples, indicating that the samples were not cytotoxic. For the determination of cell viability in ThP-1, the micellar phases were diluted in a ratio of 1:250, given the longer exposure time of the anti-inflammatory experiment (18 h). Results indicated that the diluted samples were not cytotoxic on the ThP-1 cell culture (Appendix A), as cell viability exceeded 100% in all cases. The increase in cell viability over 100% (approximately 120%) may be attributed to inconsistencies in the absorbance of MTT (a compound derived from tetrazolium salts) that did not necessarily correspond to a real increase in viability. Instead, this discrepancy could be due to possible interferences stemming from the composition of the samples, leading to a disparity between the actual count of viable cells and the count determined by the cell counting kit [27].

#### 3.4.2. CaCo-2 Cell Permeability Assay

A well-established adsorption model of Caco-2 cell monolayers on Transwell permeable supports was used for the evaluation of the degree of cell permeability of the curcumin present in the micellar phases of the digested unstructured oils (LCur, SCur, ACur) and OGs (OGLCur, OGSCur, OGACur). 

Table 3 shows the percentage of curcumin transported from the AP to the BL chamber after 1 h and 2 h of incubation with the micellar phases of digested unstructured oils and OGs. The quantity of curcumin passing through the Caco-2 cell monolayer increased with exposure time, and the percentage of curcumin in the BL compartment after 2 h was three-fold higher than that at 1 h in all the samples. Furthermore, the transfer of curcumin to the BL chamber was significantly higher for the micelle-solubilized curcumin from the digested OG (OGLCur, OGSCur, and OGACur) than from the digested unstructured oils (LCur, SCur, ACur), after both 1 and 2 h of incubation.

The apparent permeability coefficients (P_app_) of micelle-solubilized curcumin, obtained after the digestion of unstructured oils (LCur, SCur, ACur) and OGs (OGLCur, OGSCur, OGACur) in the AP-BL direction, are shown in Table 3. A rapid permeation of curcumin was observed in all the samples, with values ranging from 1.6 to 2.3 × 10^−5^ cm/s, indicating a fast absorption rate in vivo [40]. These P_app_ values were higher than those reported for DMSO-solubilized curcumin (0.05–7.10 × 10^−6^ cm/s; [14,41,42]) or protein-complexed curcumin (3.50 × 10^−6^ cm/s; [14]), possibly due to the micellation of curcumin, enhancing the cellular uptake of curcumin and BL secretion [14,41]. Yu and Huang [14] reported a permeation rate of 1.09 × 10^−5^ cm/s for curcumin encapsulated in bile-fatty acid mixed micelles, generated by the lipid digestion of monostearin curcumin OG. Although these values are slightly lower than those reported in this study, differences in the experimental conditions may account for the variation. Moreover, the P_app_ of curcumin in the digested OGs (2.1–2.3 × 10^−5^ cm/s) was slightly (*p* ≤ 0.05) higher than in the digested unstructured oils (1.6–1.8 × 10^−5^ cm/s), resulting in a higher flux and transport of curcumin through the intestinal monolayer. While the curcumin released during digestion was similar in both unstructured oils and OGs, lipolysis was higher in unstructured oils (~10%), potentially leading to increased micelle formation and lower curcumin content in micelles, consequently decreasing the P_app_ of curcumin. In this context, Yi et al. [39] reported a positive correlation between the β-carotene content in micelles and cellular uptake in Caco-2.

#### 3.4.3. Immune Stimulation Assay

As shown in Figure 3, the stimulation of ThP-1 cells with LPS for their differentiation into M1 macrophages led to an increased release of TNF-α (386.7 pg/mL, Figure 3a and 207.1 pg/mL, Figure 3b), serving as positive controls for the two assays. Treatment with the micellar phases of digested OGs significantly decreased (*p* ≤ 0.05) TNF-α production compared to the positive control, both before (Figure 3a) and after (Figure 3b) ThP-1 cell stimulation with LPS. Therefore, micelle-solubilized curcumin was able to prevent the initiation of the inflammatory process (Figure 3a) and also to interfere during the development of the inflammatory response once the cellular cascade was initiated (Figure 3b), exhibiting a potent anti-inflammatory effect. Furthermore, several fatty acids present in edible oils, such as α-linolenic acid or oleic acid, participating in the micelle structures, may also contribute to the anti-inflammatory activity observed in the micellar fraction of the digested OGs [43,44]. No significant differences (*p* > 0.05) were observed among the samples (OGLCur, OGSCur, OGACur) in the inhibition of TNF-α production, which may be attributed to a similar load of curcuminoids in all the micellar phases, given the consistent bioaccessibility values found in the three OGs (Table 2). Bisht et al. [45] also studied the anti-inflammatory potential of curcumin in macrophages differentiated from LPS-treated ThP-1 cells by measuring the amount of TNF-α, reporting that curcumin effectively reduced the TNF-α production, aligning with the results obtained in this study. Curcumin has been reported to alleviate various inflammatory processes by regulating inflammatory signaling pathways and inhibiting the production of several inflammatory mediators [46]. To the best of our knowledge, the anti-inflammatory properties of digested OGs with curcumin have not been evaluated before. These results suggest that OGs with curcumin could be promising for modulating inflammatory disorders.

## 4. Conclusions

Beeswax-based OGs, with a mechanical strength similar to pork backfat, were formulated using avocado, sunflower, and linseed oils and were evaluated as delivery vehicles for curcuminoids. The oil type used in the OG formulation had no significant impact on the bioaccessibility or intestinal absorption of curcuminoids. All the OG formulations exhibited high bioaccessibility values (55.9–61.5%) and permeation rates (2.1–2.3 × 10^−5^ cm/s) for total curcuminoids. However, the oxidative stability of the OG matrices and the degradation rate of curcumin were related to the degree of oil unsaturation. Oils with lower unsaturation displayed higher oxidative stability and degradation rates of curcumin. Hence, oil selection is a crucial factor in OG design. While the oil structuring with beeswax did not affect the bioaccessibility of individual or total curcuminoids, it reduced the extent of lipolysis (~10%) and increased the intestinal absorption of curcuminoids, evidenced by the increased flux and transport of curcumin through the intestinal monolayer in OGs. In summary, beeswax-based OGs may not only serve as an ingredient with gel-like behavior for replacing animal fat in various food applications but can also act as promising oral delivery systems for curcuminoids. Additionally, they protect unsaturated fatty acid rich-oils against lipid oxidation and provide OGs with potent anti-inflammatory capacity.

## Figures and Tables

**Figure 1 foods-13-00373-f001:**
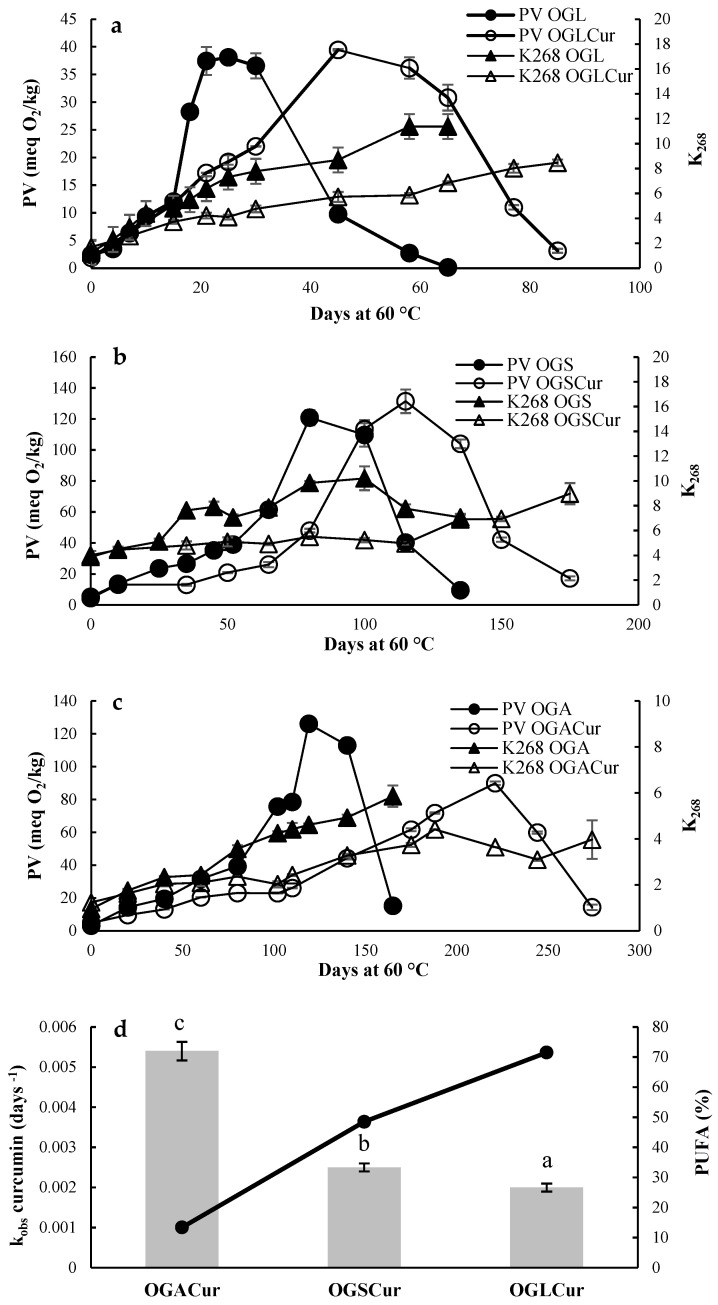
The evolution of the peroxide value (PV) and K_268_ values for OGL and OGLCur (**a**), OGS and OGSCur (**b**), and OGA and OGACur (**c**) under accelerated oxidation conditions at 60 °C. The constant curcumin degradation rate (k) in OGLCur, OGSCur, and OGACur (**d**). Different letters (a–c) denote statistically significant differences (*p* ≤ 0.05) among samples.

**Figure 2 foods-13-00373-f002:**
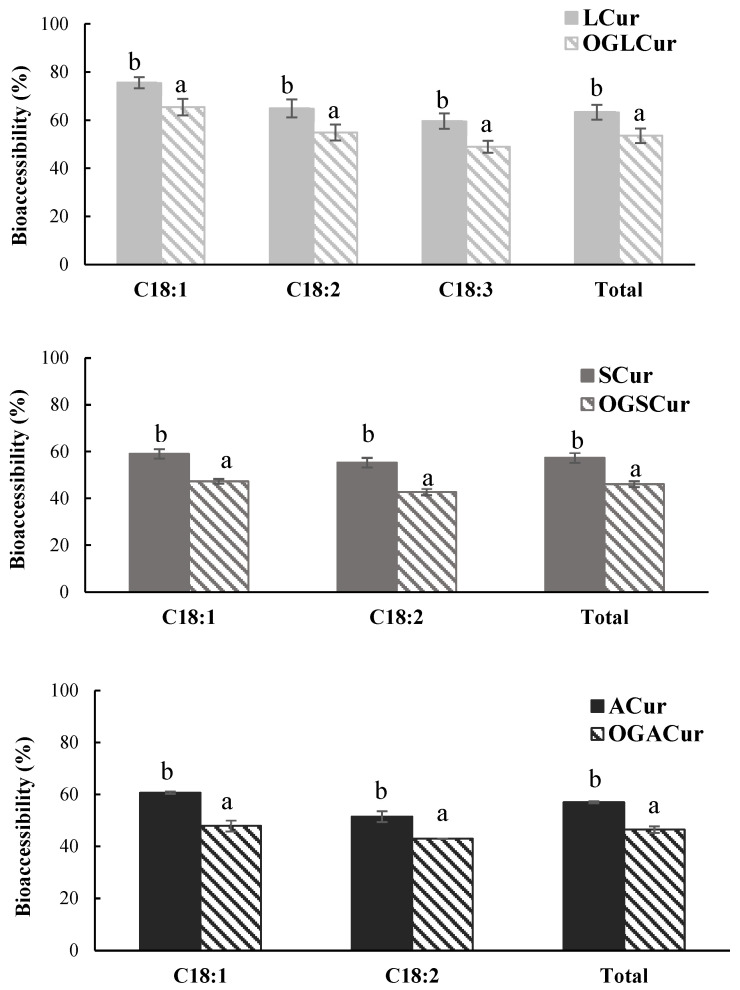
The bioaccessibility of total FFAs and major FFAs (%) at the end of the intestinal phase of the in vitro digestion of unstructured oils (LCur, SCur, ACur) and OGs (OGLCur, OGSCur, OGACur) with curcumin (0.2% *w*/*w*). Different letters (a–b) indicate significant differences (*p* ≤ 0.05) among samples.

**Figure 3 foods-13-00373-f003:**
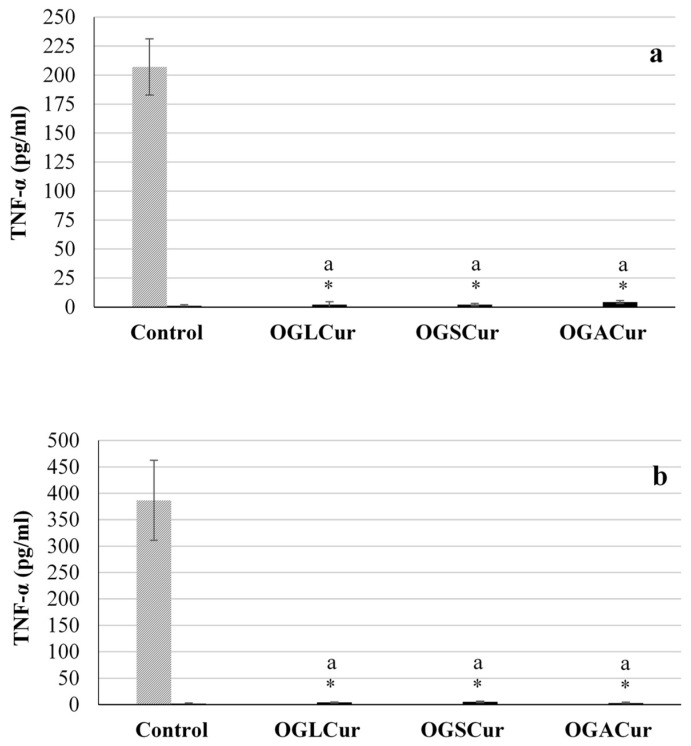
Effects of mixed micelle-solubilized curcumin from the digestion of OGs (OGLCur, OGSCur, OGACur) on TNF-α production in ThP-1 cells differentiated to macrophages. ThP-1 cells were incubated without samples (control) and with digests (OGLCur, OGSCur, OGACur) before (**a**) and after (**b**) LPS treatment for 4 h. * in samples indicates a significant difference (*p* ≤ 0.05) with respect to the positive control stimulated with LPS. Different letters (a–b) indicate significant differences (*p* ≤ 0.05) among OG samples.

**Table 1 foods-13-00373-t001:** Mechanical strength and oil binding capacity (OBC) of linseed, sunflower, and avocado beeswax OGs with curcumin.

	Mechanical Strength (N)	OBC Day 1 (%)	OBC Day 7 (%)
OGLCur	19.5 ± 1.2 ^a^	91.4 ± 1.6 ^c/y^	80.4 ± 3.7 ^b/z^
OGSCur	18.9 ± 0.4 ^a^	86.0 ± 2.6 ^a/y^	73.7 ± 1.0 ^a/z^
OGACur	18.5 ± 1.6 ^a^	88.8 ± 1.1 ^b/y^	74.5 ± 1.7 ^a/z^

OGLCur: beeswax linseed oleogel with curcumin; OGSCur: beeswax sunflower oleogel with curcumin; OGACur: beeswax avocado oleogel with curcumin; OBC: oil binding capacity. Different letters (a–c) in the same column indicate significant differences (*p* ≤ 0.05) among samples. Different letters (y–z) in the same row indicate significant differences (*p* ≤ 0.05) in OBC between day 1 and day 7.

**Table 2 foods-13-00373-t002:** The bioaccessibility (%) of curcuminoids after the in vitro digestion of OGs and unstructured oils.

	Bisdemethoxycurcumin	Demethoxycurcumin	Curcumin	Total Curcuminoids
LCur	76.1 ± 1.3 ^a/x^	67.2 ± 1.3 ^a/y^	62.0 ± 1.6 ^a/z^	63.0 ± 1.5 ^a/z^
SCur	77.6 ± 1.9 ^a/x^	67.8 ± 3.5 ^a/y^	58.2 ± 3.7 ^a/z^	61.0 ± 3.7 ^a/y,z^
ACur	75.3 ± 5.7 ^a/x^	61.9 ± 2.8 ^a/y^	57.6 ± 5.2 ^a/y^	59.8 ± 5.5 ^a/y^
OGLCur	70.2 ± 2.0 ^a/x^	61.5 ± 2.2 ^a/y^	56.8 ± 2.5 ^a/y^	58.7 ± 2.4 ^a/y^
OGSCur	78.5 ± 0.8 ^a/x^	68.2 ± 2.1 ^a/y^	59.7 ± 2.2 ^a/z^	61.5 ± 2.2 ^a/y,z^
OGACur	71.7 ± 3.1 ^a/x^	61.7 ± 2.1 ^a/y^	53.7 ± 2.6 ^a/z^	55.9 ± 2.5 ^a/y,z^

L: linseed oil; S: sunflower oil; A: avocado oil; OGL: beeswax linseed oleogel; OGS: beeswax sunflower oleogel; OGA: beeswax avocado oleogel; Cur: curcumin (0.2% *w*/*w*). Different letters (a–b) in the same column indicate significant differences (*p* ≤ 0.05) in the curcuminoid bioaccessibility due to oil structuring. Different letters (x–z) in the same row indicate significant differences (*p* ≤ 0.05) among curcuminoid bioaccessibility for the same sample.

**Table 3 foods-13-00373-t003:** Curcumin (%) that passed through the CaCo-2 cell monolayer in the AP-BL direction after 1 and 2 h of incubation. Papp values (cm/s) of micelle-solubilized curcumin were obtained after the digestion of unstructured oils and OGs.

	Curcumin (%, 1 h)	Curcumin (%, 2 h)	P_app_ (cm/s)
LCur	4.3 ± 0.1 ^a^	12.5 ± 0.5 ^a^	1.6 × 10^−5^ ± 0.2 × 10^−6 a^
SCur	4.4 ± 0.4 ^a^	13.4 ± 0.6 ^a^	1.7 × 10^−5^ ± 1.3 × 10^−6 a^
ACur	4.9 ± 0.3 ^a^	15.0 ± 0.5 ^a^	1.8 × 10^−5^ ± 1.4 × 10^−6 a^
OGLCur	5.5 ± 0.3 ^b^	17.3 ± 0.1 ^b^	2.1 × 10^−5^ ± 1.1 × 10^−6 b^
OGSCur	6.0 ± 0.1 ^b^	16.7 ± 0.1 ^b^	2.2 × 10^−5^ ± 0.4 × 10^−6 b^
OGACur	6.0 ± 0.1 ^b^	17.3 ± 0.1 ^b^	2.3 × 10^−5^ ± 0.3 × 10^−6 b^

L: linseed oil; S: sunflower oil; A: avocado oil; OGL: beeswax linseed oleogel; OGS: beeswax sunflower oleogel; OGA: beeswax avocado oleogel; Cur: curcumin (0.2% *w*/*w*). Different letters (a–b) in the same column indicate significant differences (*p* ≤ 0.05) between unstructured oils and OGs.

## Data Availability

Data is contained within the article or Appendix A.

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
