# Peer review of "Bioaccessibility, Intestinal Absorption and Anti-Inflammatory Activity of Curcuminoids Incorporated in Avocado, Sunflower, and Linseed Beeswax Oleogels"

_foods, 2024, doi:10.3390/foods13030373_

Round 1

Reviewer 1 Report

Comments and Suggestions for Authors

Hi dear Editorial board and the respected authors

This article "Bioaccessibility, intestinal absorption and anti-inflammatory activity of curcuminoids incorporated in avocado, sunflower and linseed beeswax oleogels” was revised and has a novelty and I recommend it for consideration of the following comments.

Title: If you can rewrite and make it more interesting for readers. I propose: “Bioaccessibility, intestinal absorption, and anti-inflammatory activity of curcuminoids incorporated in avocado, sunflower, and linseed beeswax oleogels”.

Abstract:

·        The type of statistical design used in this research should be mentioned.

·        K268 values?

·        What is the anti-inflammatory potential of the digested beeswax oleogels and unstructured oils in terms of inhibiting TNF-α production?

·        How do the beeswax-based oleogels, regardless of the oil type, compare to traditional fat replacers in terms of their potential as vehicles for the oral delivery of curcuminoids?

Introduction:

·         Please change the lines 86-90 to the following text: “this study aimed to provide insights into the relationship between oil type in beeswax OG formulations and the bioaccessibility, intestinal absorption, anti-inflammatory properties, and oxidative stability of curcumin. The findings will contribute to a better understanding of the potential of beeswax OG formulations as delivery vehicles for curcumin in terms of their oil type-dependent effects."

Materials:

·                      Please write materials as Company Name (City, Country), especially for chemical analysis assessment which used in the study.

Results:

·        All Tables: The alphabetical statistical letters for the means should all be modified such that the greatest number has the letter a and as the numbers go lower, letters b, c etc.

·        Table 1: in each row for the first column why did not symbol of statistical number?

·        Table 2: Please recheck totally the statistical analysis of this table because it is seem have some flaw in alphabetic letters.

Discussion:

Discussion text must grammar improve and in some cases it is very weak and maybe there is no discussion at all.

Conclusions:

Conclusion is very general, try to make it more scientific, comprehensive and concise in detail, especially.

References: It is OK.

Please explain more in the following concern the best where of the manuscript:

Beeswax-Based Olive Oil (OG) Properties and Potential Optimization

• Examines mechanical properties of OG formulated with avocado, sunflower, and linseed oils compared to pork backfat.

• Explores factors affecting bioaccessibility and intestinal absorption of curcuminoids.

• Investigates the impact of PUFA content on OG matrices' oxidative stability and curcumin degradation rate.

• Explores interactions between oil structuring with beeswax and bioaccessibility of various curcuminoids.

• Investigates potential mechanisms influencing lipolysis reduction and intestinal absorption with oil structuring.

• Explores beeswax-based OG as a gel-like ingredient for animal fat replacement in food applications.

• Quantifies and compares beeswax-based OG's lipid oxidation protection.

• Identifies specific components contributing to beeswax-based OG's potent anti-inflammatory capacity.

Comments on the Quality of English Language

The article has many flaws in express and concept of English, it is suggested to be revised in a scientific and native way.

Author Response

Dear Reviewer,

We are sending all the files concerning to the revised version of our paper titled “Bioaccessibility, intestinal absorption and anti-inflammatory activity of curcuminoids incorporated in avocado, sunflower, and linseed beeswax oleogels”, by Patricia Ramírez-Carrasco, Ailén Alemán, Estefanía González, M. Carmen Gómez-Guillén, Paz Robert, and Begoña Giménez, to be submitted for publication in Foods.

The responses to your comments are in red colour in the manuscript. Thank you very much for your suggestions.

Title: If you can rewrite and make it more interesting for readers. I propose: “Bioaccessibility, intestinal absorption, and anti-inflammatory activity of curcuminoids incorporated in avocado, sunflower, and linseed beeswax oleogels”.

The title has been modified as suggested by the referee.

Abstract:

The type of statistical design used in this research should be mentioned.

The statistical design has been mentioned in the abstract as suggested by the referee.

K268 values?

K268 values have been defined in the abstract.

What is the anti-inflammatory potential of the digested beeswax oleogels and unstructured oils in terms of inhibiting TNF-α production?

TNF-α is one of the key cytokines mediating the inflammatory response. This cytokine is usually used to evaluate the anti-inflammatory potential of bioactive compounds. The sentence (lines 24-28) has been completed for a better understanding:

The intestinal absorption, evaluated in Caco-2 cell monolayers, was higher for the micelle-solubilized curcumin from the digested OG than from unstructured oils, and showed high anti-inflammatory potential by inhibiting the tumor necrosis factor-α (TNF-α) production compared to the positive control, both before and after stimulation of ThP-1 cells with LPS.

How do the beeswax-based oleogels, regardless of the oil type, compare to traditional fat replacers in terms of their potential as vehicles for the oral delivery of curcuminoids?

The performance of traditional fats as vehicles for the oral delivery of curcuminoids has not been studied and compared with OG, since the objective is the design of traditional fat replacers to improve the fatty acid profile of foods where these oleogels can be incorporated, such as meat products (Ramírez-Carrasco et al., 2020).

Introduction:

Please change the lines 86-90 to the following text: “this study aimed to provide insights into the relationship between oil type in beeswax OG formulations and the bioaccessibility, intestinal absorption, anti-inflammatory properties, and oxidative stability of curcumin. The findings will contribute to a better understanding of the potential of beeswax OG formulations as delivery vehicles for curcumin in terms of their oil type-dependent effects."

The sentence has been changed, as suggested by the referee (lines 91-95).

Materials:

Please write materials as Company Name (City, Country), especially for chemical analysis assessment which used in the study.

The grade and company name, city and country of the reagents used has been added in the Materials section. Furthermore, the culture media used have been also added to the manuscript, indicating company name, city and country, as suggested by the referee (lines 107-109; lines 113-116).

Results:

All Tables: The alphabetical statistical letters for the means should all be modified such that the greatest number has the letter a and as the numbers go lower, letters b, c etc.

The letter assignment has been modified as suggested by the referee.

Table 1: in each row for the first column why did not symbol of statistical number?

Because the mechanical strength is not compared with the OBC values of each OG, but the mechanical strength values among the three OG.

Table 2: Please recheck totally the statistical analysis of this table because it is seem have some flaw in alphabetic letters.

The letter assignment of Table 2 is correct. The effect of the oil structuring was analyzed, and therefore, all the results of liquid oils were compared with all the results of OG for total curcuminoids and every curcuminoid. Furthermore, the effect of the oil type on the bioaccessibility of curcumin and total curcuminoids from OG was also analyzed, but this analysis was indicated in the text in order to simplify the statistical analysis of Table 2 with the following sentence (lines 501-504): The oil type used in the OG formulation did not exert any significant effect on the bioaccessibility of curcumin or total curcuminoids, and all the OG showed similar (p>0.05) values, ranging from 53.7% to 59.7% for curcumin and 55.9% to 61.5% for total curcuminoids.

Discussion:

Discussion text must grammar improve and in some cases it is very weak and maybe there is no discussion at all.

Discussion has been enriched according to the suggestions made below by the referee.

Conclusions:

Conclusion is very general, try to make it more scientific, comprehensive and concise in detail, especially.

Conclusions has been rewritten as suggested by the referee.

References: It is OK.

Please explain more in the following concern the best where of the manuscript:

Beeswax-Based Olive Oil (OG) Properties and Potential Optimization

Olive oil was not used to formulate OG in this study.

  • Examines mechanical properties of OG formulated with avocado, sunflower, and linseed oils compared to pork backfat.

The term “mechanical properties” has been replaced by “mechanical strength” throughout the manuscript.

  • Explores factors affecting bioaccessibility and intestinal absorption of curcuminoids.

The main factors affecting bioaccessibility and intestinal absorption of curcuminoids are explained in the manuscript (lines 66-68): “However, its practical application is constrained due to its low water solubility, chemical instability and fast metabolism, significantly impacting its bioaccessibility and oral bioavailability”.

Nevertheless, for a better understanding this sentence has been completed as follows (lines 68-71): “To address these limitations, enhancing the solubility of curcumin through the development of delivery systems, such as lipid-based vehicles, is one of the strategies which has received increasing attention in recent years (Yu et al., 2015).”

  • Investigates the impact of PUFA content on OG matrices' oxidative stability and curcumin degradation rate.

The impact of PUFA content on OG oxidative stability was investigated and described in the manuscript (lines 371-381). The evolution of PV was in agreement with the degree of oil unsaturation, and the higher the oil unsaturation the shorter the storage time at which the OG reached the maximum PV. Furthermore, the maximum PV value was also related with the degree of oil unsaturation, and the linseed OG reached lower maximum PV values than the sunflower and avocado OGs, suggesting that lower PV is related to highly unsaturated systems, where the peroxide decomposition is faster than the formation and therefore, the maximum PV achieved is lower. These results are in agreement with other studies that have been quoted in the manuscript (Lim et al., 2017; Hwang et al., 2018).

The impact of PUFA content on curcumin degradation rate is shown in Figure 1d, where degradation rate constant of curcumin is higher in oils with low degree of unsaturation, such as avocado oil, but lower in oils with high degree of unsaturation such as sunflower and specially linseed oil. For a better understanding, the sentence in lines 401-404 has been completed as following: These results suggest that the degree of oil unsaturation may influence the degradation rate of curcumin when incorporated into lipophilic matrices, with higher degradation rates in oils with low degree of unsaturation.

  • Explores interactions between oil structuring with beeswax and bioaccessibility of various curcuminoids.

Although the type of structuring agent may significantly affect the bioaccessibility of curcuminoids incorporated in OG, both increasing or decreasing the bioaccessibility values (Calligaris et al., 2020; Li et al., 2019), oil structuring with beeswax did not interfere with the bioaccessibility of any curcuminoid (total curcuminoids, bisdemethoxycurcumin, demethoxycurcumin and curcumin) in this study. This was explained in the manuscript (lines 506-519) and indicated in the statistical analysis of Table 2.

  • Investigates potential mechanisms influencing lipolysis reduction and intestinal absorption with oil structuring.

The mechanisms influencing lipolysis with oil structuring have been rewritten in lines 426-439 for a better understanding: “The type and gelation mechanism of the oil structuring agent employed in the OG formulation, the mechanical strength of the OG and the presence of several components in the matrix that may directly interfere with the digestion process can influence the extent of lipolysis. A sufficiently structured OG network can impede the access of lipases and bile salts to the liquid oil trapped in the OG matrix, thereby preventing lipid hydrolysis and emulsification and, consequently, reducing lipolysis. OG with high mechanical strength have been associated with a lower extent of lipolysis in some studies where ethylcellulose or phytosterols OG were subjected to in vitro digestion [18,37,38]. This is attributed to the fact that harder OG are more resistant to mechanical breakdown during digestion, leading to a slower released of oil that remain inaccessible to lipases. However, this relation between mechanical strength and extent of lipolysis is not universally consistent. Ashkar et al. [37] reported an inverse relationship between OG hardness and the extent of lipolysis, which was attributed to the contribution of the oil structuring agent (monoglycerides and diglycerides) in micelle formation during digestion.”

  • Explores beeswax-based OG as a gel-like ingredient for animal fat replacement in food applications.

The focus of this study was to provide insights into the relationship between oil type in beeswax OG formulations and the bioaccessibility, intestinal absorption, anti-inflammatory properties, and oxidative stability of curcumin. The potential of OG as fat replacers in several foods has been addressed in the introduction section (lines 44-55). Furthermore, authors have successfully explored beeswax-based linseed OG as gel-like ingredients for pork backfat replacement in spreadable meat products (pâté) in a previous study (Ramírez-Carrasco et al., 2020), and currently, authors are working in using sunflower and avocado OG as fat replacers in other meat products, the results of which will be published in other paper.

  • Quantifies and compares beeswax-based OG's lipid oxidation protection.

The effect of oil structuring on lipid oxidation protection was not determined, since the storage temperature was above the melting temperature of the OGs. In spite of this, the length of the oxidation stability study performed in this work was 85, 180 and 280 days for linseed, sunflower and avocado OGs, respectively, to cover the complete curves of formation and degradation of peroxides as well as the curcumin degradation. As indicated in the text (lines 410-414), both lipid oxidation and curcumin degradation would be expected to slow down at storage temperatures under the melting point due to the formation of the crystalline network immobilizing the oil, which in turn reduces the oxygen diffusion coefficient and restricts the movement of reactants.

  • Identifies specific components contributing to beeswax-based OG's potent anti-inflammatory capacity.

The anti-inflammatory capacity was evaluated in the micellar fraction obtained after the in vitro digestion of OGs, where curcumin was solubilized. These micelles are mainly composed of bile salts, free fatty acids and monoglycerides coming from the lipolysis of OGs, with the curcumin released and solubilized in their hydrophobic core. Therefore, the anti-inflammatory activity found in the micellar fraction was mainly attributed to the micelle-solubilized curcumin. In spite of this, it is known that several fatty acids present in edible oils, such as α-linolenic acid or oleic acid, participating in the micelle structures, may also contribute to the anti-inflammatory activity observed in the micellar fraction of the digested OG (Kim et al., 2014; Santa-María et al., 2023). This has been incorporated in the manuscript (lines 597-600).

Furthermore, some minor components within edible oils may also contribute to the anti-inflammatory activity of the digested OG, such as tocopherols, phytosterols and phenolic compounds (Zeb, 2021; Adeleke & Babalola, 2020; Elosaily et al., 2021), but these compounds were not determined in the micellar fraction from the digested OG.

Comments on the Quality of English Language

The article has many flaws in express and concept of English, it is suggested to be revised in a scientific and native way.

English has been revised throughout the manuscript.

Reviewer 2 Report

Comments and Suggestions for Authors

The potentials of many bioactive molecules for health benefits are not fully exploited due to constraints such as poor solubility in commonly used dispersing media, less than desirable stability, photo degradation, lower levels of bio accessibility/bioavailability. Several innovative approaches are taken to overcome technical problems for full utilization of nutraceutical molecules. Some of the approaches used for this purpose include use of emulsifiers, micelles, liposomes, nanoparticles, nanogels. Many investigators have used different core materials and delivery systems in their work. Oleo gels have also received attention to improve the efficacy of nutraceuticals. In the present manuscript, the authors have used beeswax  based oleo gels and utilized its mechanical properties to replace animal fats like pork back fat in food applications. The oleo gels were prepared to contain vegetable oils like avocado, sunflower and linseed oils. These oils differ in their fatty acid composition as well as degree of unsaturation of fatty acids . The prepared oleo gels were tested as delivery vehicles for a model compound like curcuminoids . Curcumin is receiving considerable attention for its medicinal and pharmaceutical  properties. However stability, photooxidation ,low bioavailability and rapid metabolism in vivo often restrict its utility. In the present work, it is shown that the entrapped curcuminoids in oleo gels  have better oxidative stability and enhanced bio accessibility and bioavailability. The type of oils provided in the oleo gels did however not affected this observation. Curcumin delivered in gels exhibited better anti-inflammatory activity which was monitored by measuring TNF alpha levels in activated cells ,Based on these results, the authors concluded that oleo gels with vegetable oils could be used in place of animal fats in food applications. This may help in acceptance of oleo gels in foods by vegetarian sections of population. Oral delivery of curcuminoids through gels is a promising area for extending its utilization for health benefits.

Clarification to provide:

What are the foods which authors would like to try in future for utilizing oleo gels with high value nutraceuticals?

Comments on the Quality of English Language

Minor editing is required

Author Response

Dear Reviewer,

We are sending all the files concerning to the revised version of our paper titled “Bioaccessibility, intestinal absorption and anti-inflammatory activity of curcuminoids incorporated in avocado, sunflower, and linseed beeswax oleogels”, by Patricia Ramírez-Carrasco, Ailén Alemán, Estefanía González, M. Carmen Gómez-Guillén, Paz Robert, and Begoña Giménez, to be submitted for publication in Foods.

Thank you very much for your suggestions.

- What are the foods which authors would like to try in future for utilizing oleo gels with high value nutraceuticals?

We have successfully used beeswax-based linseed OGs as gel-like ingredients for pork backfat replacement in spreadable meat products (pâté) in a previous study (Ramírez-Carrasco et al., 2020), and currently, we are working in using sunflower and avocado OG as fat replacers in other meat products, such as sausages, the results of which will be published in other paper.

Reviewer 3 Report

Comments and Suggestions for Authors

In the manuscript by Ramírez-Carrasco et al. entitled »Bioaccessibility, intestinal absorption and anti-inflammatory activity of curcuminoids incorporated in avocado, sunflower and linseed beeswax oleogels« the authors investigate the possibility of using beeswax-based oleogels as a vehicle for the oral administration of curcuminoids. Their conclusions that this type of oleogels could not only be a promising substitute for animal fats in various food applications, but also be suitable as a delivery system for the phenolic compounds of the curcuminoid family, seem to be quite well supported by the study described in the manuscript.

Therefore, I propose minor improvements (see below):

-             While the abbreviation GRAS (line 38) should be familiar to most readers of the journal Foods, this is less true for some other abbreviations (e.g. TNF-a (line 26)). However, to ensure that the meaning of all abbreviations is clear to all readers, I suggest that all abbreviations used are explained in the text at the point where they first appear. I suggest that this also applies to abbreviations in the abstract (e.g. PUFA), or alternatively that no abbreviations are used in the abstract.

-             I suggest that a graph showing a typical HPLC chromatogram of curcuminoids be included in the Supplementary Materials

Comments on the Quality of English Language

Some minor improvements of English language are possible.

Author Response

Dear Reviewer,

We are sending all the files concerning to the revised version of our paper titled “Bioaccessibility, intestinal absorption and anti-inflammatory activity of curcuminoids incorporated in avocado, sunflower, and linseed beeswax oleogels”, by Patricia Ramírez-Carrasco, Ailén Alemán, Estefanía González, M. Carmen Gómez-Guillén, Paz Robert, and Begoña Giménez, to be submitted for publication in Foods.

The response to your comments are in blue in the manuscript. Thank you very much for your suggestions.

- While the abbreviation GRAS (line 38) should be familiar to most readers of the journal Foods, this is less true for some other abbreviations (e.g. TNF-a (line 26)). However, to ensure that the meaning of all abbreviations is clear to all readers, I suggest that all abbreviations used are explained in the text at the point where they first appear. I suggest that this also applies to abbreviations in the abstract (e.g. PUFA), or alternatively that no abbreviations are used in the abstract.

All the abbreviations have been revised throughout the manuscript, as suggested by the referee.

- I suggest that a graph showing a typical HPLC chromatogram of curcuminoids be included in the Supplementary Materials

A typical HPLC chromatogram of curcuminoids have been included as supplementary material.

Reviewer 4 Report

Comments and Suggestions for Authors

The authors of the manuscript "Bioaccessibility, intestinal absorption and anti-inflammatory activity of curcuminoids incorporated in avocado, sunflower and linseed beeswax oleogels" used beeswax oleogels to prepare a fat corresponding to animal fat - lard. The test results indicate the possibility of producing vegetable fats that are analogues of solid animal fats. The purpose of the research is directed in an interesting and correct manner. However, the presented manuscript has some issues that require refinement.

1. Why were three unrefined cold-pressed oils selected for the experiment, but one of them, sunflower oil, was refined. However, such a selection introduces some research inaccuracies.

2. In chapter 2.2.1, I would briefly provide the concentrations of beeswax used and not only refer to previous works

3. In chapter 2.2.5.2, the authors use a reference to chapter 2.2.4.2 which is not included in the text.

4. In chapter 2.2.5.3. please provide the incubation temperature

Author Response

Dear Reviewer,

We are sending all the files concerning to the revised version of our paper titled “Bioaccessibility, intestinal absorption and anti-inflammatory activity of curcuminoids incorporated in avocado, sunflower, and linseed beeswax oleogels”, by Patricia Ramírez-Carrasco, Ailén Alemán, Estefanía González, M. Carmen Gómez-Guillén, Paz Robert, and Begoña Giménez, to be submitted for publication in Foods.

The response to your comments are in green in the manuscript. Thank you very much for your suggestions.

  1. Why were three unrefined cold-pressed oils selected for the experiment, but one of them, sunflower oil, was refined. However, such a selection introduces some research inaccuracies.

Refined sunflower oil was used in this study because unrefined cold-pressed sunflower oil is not produced in Chile. However, the possible implications in OG formulation have been addressed in chapter 3.1.

  1. In chapter 2.2.1, I would briefly provide the concentrations of beeswax used and not only refer to previous works

Beeswax concentrations have been provided in chapter 2.2.1.

  1. In chapter 2.2.5.2, the authors use a reference to chapter 2.2.4.2 which is not included in the text.

Chapter 2.2.4.2. has been replaced by 2.2.4.1.

  1. In chapter 2.2.5.3. please provide the incubation temperature

The incubation conditions have been provided (37 °C and 5% CO2).
